# Clarifying Microbial Nitrous Oxide Reduction under Aerobic Conditions: Tolerant, Intolerant, and Sensitive

Zhiyue Wang,[a,b] Nisha Vishwanathan,[c] Sophie Kowaliczko,[c] Satoshi Ishii[c,d]

[a]Department of Civil and Environmental Engineering, University of Hawai'i, Honolulu, Hawai'i, USA
[b]Water Resources Research Center, University of Hawai'i, Honolulu, Hawai'i, USA
[c]BioTechnology Institute, University of Minnesota, St. Paul, Minnesota, USA
[d]Department of Soil, Water, and Climate, University of Minnesota, St. Paul, Minnesota, USA

**ABSTRACT** One of the major challenges for the bioremediation application of microbial nitrous oxide ($N_2O$) reduction is its oxygen sensitivity. While a few strains were reported capable of reducing $N_2O$ under aerobic conditions, the $N_2O$ reduction kinetics of phylogenetically diverse $N_2O$ reducers are not well understood. Here, we analyzed and compared the kinetics of clade I and clade II $N_2O$-reducing bacteria in the presence or absence of oxygen ($O_2$) by using a whole-cell assay with $N_2O$ and $O_2$ microsensors. Among the seven strains tested, $N_2O$ reduction of *Stutzerimonas stutzeri* TR2 and ZoBell was not inhibited by oxygen (i.e., oxygen tolerant). *Paracoccus denitrificans*, *Azospirillum brasilense*, and *Gemmatimonas aurantiaca* reduced $N_2O$ in the presence of $O_2$ but slower than in the absence of $O_2$ (i.e., oxygen sensitive). $N_2O$ reduction of *Pseudomonas aeruginosa* and *Dechloromonas aromatica* did not occur when $O_2$ was present (i.e., oxygen intolerant). Amino acid sequences and predicted structures of NosZ were highly similar among these strains, whereas oxygen-tolerant $N_2O$ reducers had higher oxygen consumption rates. The results suggest that the mechanism of $O_2$ tolerance is not directly related to NosZ structure but is rather related to the scavenging of $O_2$ in the cells and/or accessory proteins encoded by the *nos* cluster.

**IMPORTANCE** Some bacteria can reduce $N_2O$ in the presence of $O_2$, whereas others cannot. It is unclear whether this trait of aerobic $N_2O$ reduction is related to the phylogeny and structure of $N_2O$ reductase. The understanding of aerobic $N_2O$ reduction is critical for guiding emission control, due to the common concurrence of $N_2O$ and $O_2$ in natural and engineered systems. This study provided the $N_2O$ reduction kinetics of various bacteria under aerobic and anaerobic conditions and classified the bacteria into oxygen-tolerant, -sensitive, and -intolerant $N_2O$ reducers. Oxygen-tolerant $N_2O$ reducers rapidly consumed $O_2$, which could help maintain the low $O_2$ concentration in the cells and keep their $N_2O$ reductase active. These findings are important and useful when selecting $N_2O$ reducers for bioremediation applications.

**KEYWORDS** nitrous oxide reduction, oxygen sensitivity, microsensor, kinetics, enzyme kinetics, microsensors

Nitrous oxide ($N_2O$) is a potent greenhouse gas and a stratospheric ozone layer destructor (1). The use of microbial $N_2O$ reduction has a potential to mitigate $N_2O$ emissions (2, 3). This reaction is catalyzed by nitrous oxide reductase ($N_2OR$) encoded by the *nos* cluster (4). $N_2OR$ is the only known enzyme so far capable of biologically reducing $N_2O$ to $N_2$ and is carried by both denitrifying and nondenitrifying microorganisms (5).

$N_2OR$ is generally believed to be sensitive to oxygen ($O_2$), which may limit the bioremediation application of $N_2OR$ in a standard aerobic environment. Exposure to oxygen may change the configuration of the copper-based catalytic sites and inactivate $N_2OR$ (6). Such inactivation could potentially protect the enzyme from irreversible damage

Address correspondence to Satoshi Ishii, ishi0040@umn.edu.

The authors declare no conflict of interest.

and the production of reactive oxygen radicals upon transient exposure to oxygen (7). This process could also contribute to the sensitivity of N$_2$OR to oxygen at the enzyme level. In addition to the effect on the enzyme itself, O$_2$ can also influence the transcription of the *nos* cluster. The O$_2$-sensing transcription regulators, such as FNR and NNR, as well as small RNA, can suppress the transcription of *nos* (8, 9).

While the impact of O$_2$ on microbial N$_2$O reduction has been well documented, some denitrifying bacterial strains have been reported to reduce N$_2$O in the presence of O$_2$ (i.e., aerobic N$_2$O reduction) (10, 11). However, the ecophysiology of aerobic N$_2$O reduction remains largely unclear. Questions that remained unanswered include whether the O$_2$ sensitivity of N$_2$OR is related to their structure and how widely aerobic N$_2$O reducers occur in the N$_2$OR phylogeny.

There are two distinct clades (clade I and II) for *nosZ*, which is the key functional gene of N$_2$OR (12). Genomic differences between the two clades are associated with *nos* cluster organization, the translocation pathway, and co-occurrence with other denitrifying genes (13). Several studies have reported the physiological differences between the two clades. Yoon et al. (14) report that clade II bacteria (*Dechloromonas aromatica* and *Anaeromyxobacter dehalogenans*) showed high affinities to N$_2$O but lower maximum reduction rates than those of clade I bacteria (*Stutzerimonas stutzeri*, formerly known as *Pseudomonas stutzeri* [15], and *Shewanella loihica*). In contrast, Suenaga et al. (3) found that the N$_2$O reduction biokinetics could not be used to distinguish the clade I bacteria (*S. stutzeri* and *Paracoccus denitrificans*) and clade II bacteria studied (*Azospira* spp.). Nevertheless, it is still unclear how clade I and II N$_2$O reducers behave in the presence of O$_2$.

Therefore, the objectives of this study were to (i) characterize the oxygen sensitivity of various N$_2$O reducing bacteria, (ii) classify N$_2$OR based on their oxygen sensitivity, and (iii) examine the relationships between N$_2$OR oxygen sensitivity, *nosZ* phylogeny (clade I versus clade II), and the predicted N$_2$OR structures.

## RESULTS

**Michaelis-Menten kinetics of aerobic and anaerobic N$_2$O reduction.** By fitting the N$_2$O reduction results normalized by the optical density (OD) at 600 nm wavelength to the Michaelis-Menten model, we obtained the maximum rate ($V_{max}$) and Michaelis constant ($K_m$) values for various N$_2$O-reducing strains under aerobic and anaerobic conditions. A wide range of $V_{max}$ for nitrous oxide reduction rates was observed. Under anaerobic conditions, bacteria with clade I N$_2$OR generally exhibited faster N$_2$O reduction than those with clade II N$_2$OR (Fig. 1 and 2). Under anaerobic conditions, *S. stutzeri* TR2 (clade I N$_2$OR) (Fig. 1-B1) had the highest $V_{max}$ (8.37 $\pm$ 0.81 $\mu$M/s/OD), whereas *G. aurantiaca* T-27 (clade II N$_2$OR) (Fig. 2-C1) had the lowest $V_{max}$ (0.13 $\pm$ 0.02 $\mu$M/s/OD). This general trend in kinetics, however, may not extend beyond the studied strains, especially given the diversity of microorganisms harboring clade II NosZ (12).

The ability to reduce N$_2$O in the presence of O$_2$ varied by strain, and there was no overall trend between the tested strains with clade I and II N$_2$OR. For example, *S. stutzeri* TR2 (Fig. 1-B2) reduced N$_2$O under aerobic conditions with the $V_{max}$ of 6.65 $\pm$ 0.37 $\mu$M/s/OD, whereas *Pseudomonas aeruginosa* PAO1 (clade I N$_2$OR) (Fig. 2-A1) could not reduce N$_2$O in the presence of O$_2$. *D. aromatica* RCB (clade II N$_2$OR) (Fig. 2-B1) also could not reduce N$_2$O in the presence of O$_2$. In contrast, *Gemmatimonas aurantiaca* T-27 (clade II N$_2$OR) (Fig. 2-C2) exhibited a very slow N$_2$O reduction rate under aerobic conditions, which increased once O$_2$ was depleted. *Azospirillum brasilense* Sp7 (clade I N$_2$OR) (Fig. 1-D2) reduced N$_2$O in the presence of O$_2$ up to 180 $\mu$M; however, its $V_{max}$ could not be fitted to the Michaelis-Menten model.

The transition points from aerobic to anaerobic N$_2$O reductions (i.e., the change of the slopes between two linear rates) were clearly observed after oxygen was depleted for all tested strains, except for *G. aurantiaca* T-27. For *G. aurantiaca* T-27, the N$_2$O reduction rate gradually changed depending on the oxygen concentration (Fig. 2-C2). In order to further investigate the different oxygen inhibition kinetics observed for *G. aurantiaca*, nonlinear least square fitting with multiple variables was used to determine the inhibition constant ($K_i$). The noncompetitive inhibition model was found to best describe the changing $V_{max}$

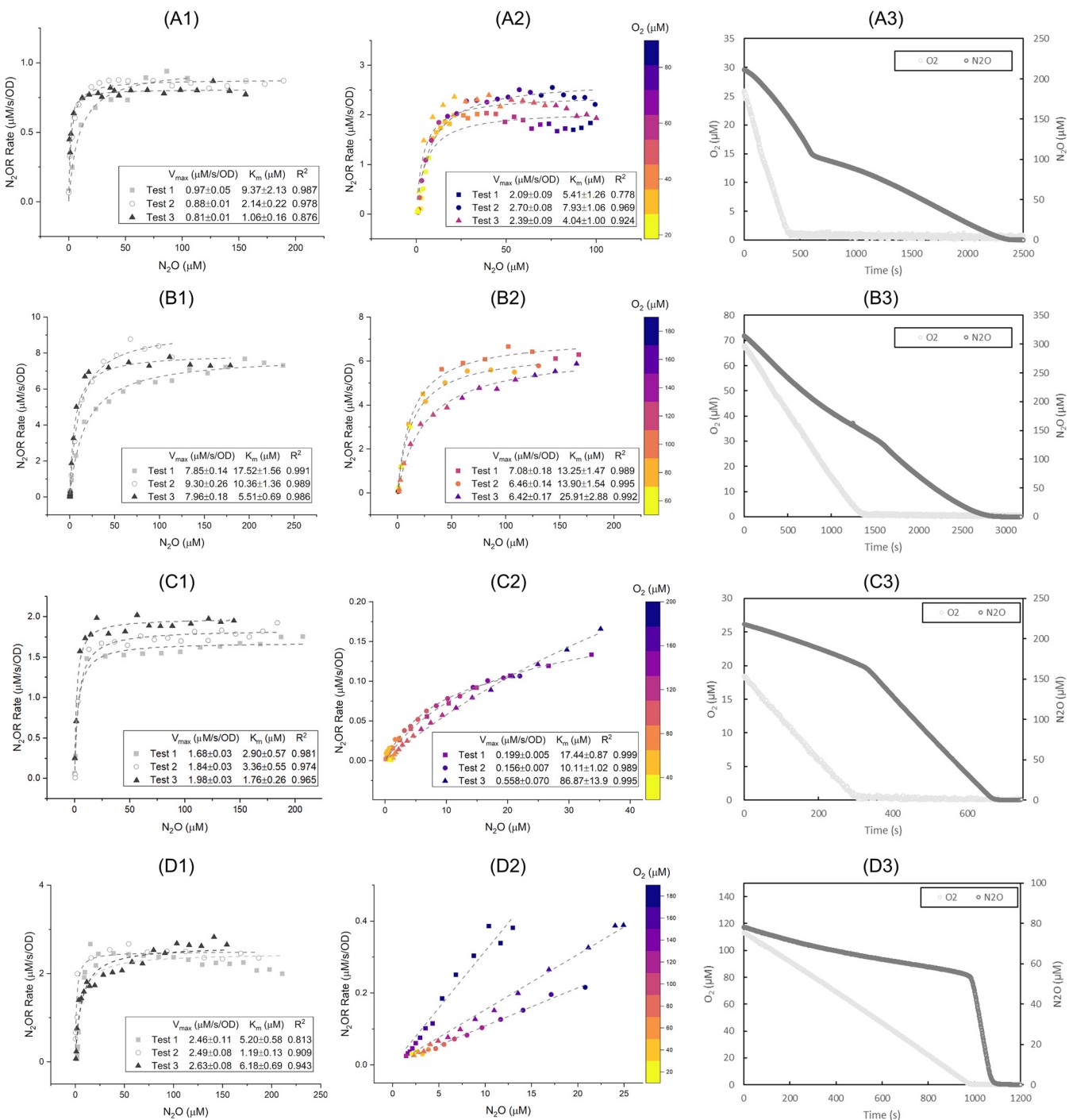

**FIG 1** Michaelis-Menten kinetics of anaerobic (1) and aerobic (2) N$_2$O reduction and transition of aerobic into anaerobic N$_2$O reduction (3) from *Stutzerimonas stutzeri* ZoBell (A), *Stutzerimonas stutzeri* TR2 (B), *Paracoccus denitrificans* JCM 21484 (C), and *Azospirillum brasilense* Sp7 (D). Curve fitting results were plotted in dashed lines.

against various O$_2$ and N$_2$O concentrations (see Fig. S1 in the supplemental material), with a $K_i$ value of 7.86 ± 1.69 $\mu$M O$_2$.

The fitted $K_m$ values for anaerobic N$_2$O reduction ranged from 1.85 ± 1.25 $\mu$M (for *D. aromatica*) to 11.14 ± 6.04 $\mu$M (for *S. stutzeri* TR2). The $K_m$ values of aerobic N$_2$O reduction for *P. denitrificans* and *S. stutzeri* TR2 and ZoBell strains did not significantly differ from those of anaerobic N$_2$O reduction (Student's *t* test, $P > 0.05$). This finding indicates that the affinity of clade I N$_2$OR tested did not change with and without the presence of O$_2$.

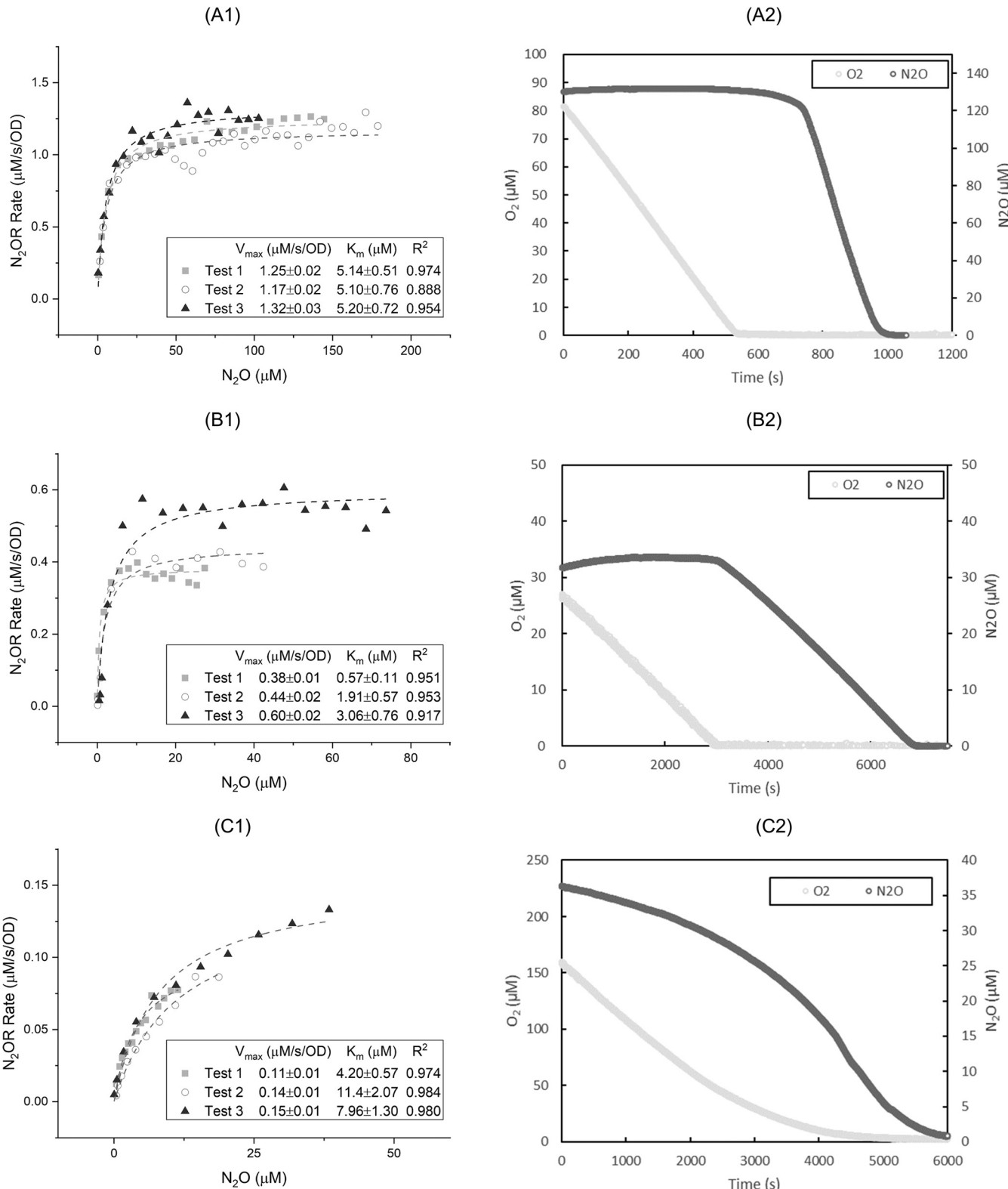

**FIG 2** Michaelis-Menten kinetics of anaerobic $N_2O$ reduction (1) and the transition of aerobic respiration into anaerobic $N_2O$ reduction (2) from *Pseudomonas aeruginosa* PAO1 (A), *Dechloromonas aromatica* RCB (B), and *Gemmatimonas aurantiaca* T-27 (C). Curve fitting results were plotted in dashed lines.

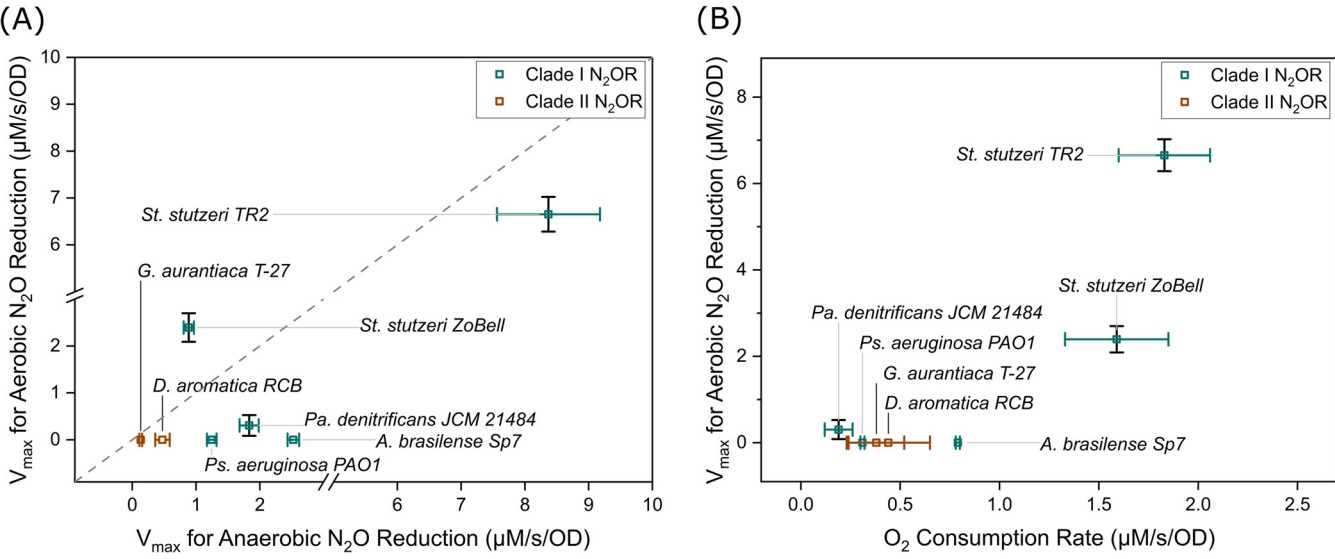

**FIG 3** $V_{max}$ for aerobic N₂O reduction rates versus $V_{max}$ for anaerobic N₂O reduction rates (A) and $V_{max}$ for O₂ consumption (B) of each studied strain.

**Classification of oxygen sensitivity of N₂O reduction.** Based on the microsensor analysis, a broad range of N₂O reduction kinetics was observed under aerobic and anaerobic conditions. As we plotted the extrapolated anaerobic and aerobic $V_{max}$ values (Fig. 3A), three distinct types of responses to oxygen were found in the studied strains, as follows: (i) strains with $V_{max}$ not affected by oxygen, including *S. stutzeri* ZoBell and TR2, are classified as oxygen tolerant; (ii) strains with much lower aerobic $V_{max}$ than anaerobic $V_{max}$, including *P. denitrificans*, *A. brasilense*, and *G. aurantiaca*, are classified as oxygen sensitive; and (iii) strains that have no N₂O reduction activity when oxygen is present, including *P. aeruginosa* and *D. aromatica*, are classified as oxygen intolerant. NosZ phylogeny seems to be not associated with the classification of oxygen sensitivity. Moreover, the half-saturation coefficients for N₂O under anaerobic and aerobic conditions agree with previously reported observations. Bacteria harboring clade II NosZ generally have lower $K_m$ values than those with clade I NosZ, suggesting differentiating ecological niches for these two groups of N₂O-reducing bacteria (14).

**NosZ amino acid sequence similarities among the strains.** The NosZ amino acid sequences of the strains studied were compared to examine whether the observed differences in oxygen sensitivity originate from the differences in the enzyme structures. Strains investigated in this study cover a variety of classes, including *Alphaproteobacteria* (*A. brasilense* and *P. denitrificans*) and *Gammaproteobacteria* (*S. stutzeri* and *P. aeruginosa*) for those having clade I NosZ and *Betaproteobacteria* (*D. aromatica*) and *Gemmatimonadetes* (*G. aurantiaca*) for those having clade II NosZ. Based on the NosZ phylogenetic analysis, clade I and clade II NosZ were clearly separated (Fig. 4), similar to the previous report (16). The two *S. stutzeri* strains, of which both showed oxygen-tolerant N₂O reduction, shared a high similarity in the NosZ amino acid sequences (92.6%) (see Fig. S4 in the supplemental material). However, *P. aeruginosa* PAO1, which showed oxygen-intolerant N₂O reduction, also has similar NosZ amino acid sequences to *S. stutzeri* (77.5% with the ZoBell strain and 79.7% with the TR2 strain). NosZ of oxygen-sensitive N₂O reducers (*P. denitrificans*, *A. brasilense*, and *G. aurantiaca*) and oxygen-intolerant N₂O reducers (*P. aeruginosa* and *D. aromatica*) were not clustered with each other. In addition, we could not identify amino acid residues that appeared specific to each of the oxygen-tolerant, -sensitive, and -intolerant groups.

Multiple sequence alignment showed that the candidate ligands of $Cu_A$ and $Cu_Z$ centers were found in all NosZ sequences (see Fig. S3 in the supplemental material). The $Cu_Z$ catalytic site contains seven histidine ligands which were all conserved in the proposed $Cu_Z$ center among clade I and clade II (Fig. S3). The candidate ligands of $Cu_A$ (two cysteines

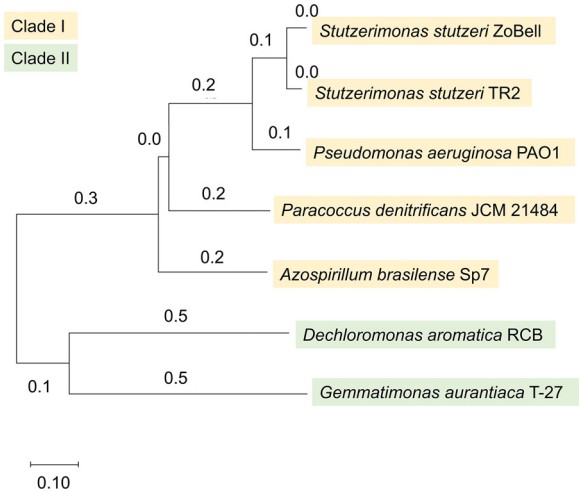

**FIG 4** Phylogenetic tree for selected NosZ sequence constructed by the neighbor-joining method.

at positions 618 and 622, two histidines at positions 583 and 626, and a methionine at position 629 in *P. stutzeri* NosZ) were also identified in all NosZ (Fig. S3).

**NosZ structural similarities.** To identify the structural differences between oxygen-tolerant, -sensitive, and -intolerant NosZ, we predicted the enzyme structures based on the NosZ sequences by using Alphafold2 (17) with the ZoBell NosZ (18) as a query structure. We obtained high-confidence NosZ structures, as evaluated based on the sequence coverage and predicted per-residue confidence measure (pLDDT) scores from AlphaFold, with conserved Cu$_A$ and Cu$_Z$ catalytic domains (see Fig. S2 in the supplemental material). Slight structural differences were seen between clade I and II NosZ as measured by the Dali Z-scores, whereas no differences were seen between NosZ structures from oxygen-tolerant, -sensitive, and -intolerant strains. The Z scores for all clade I NosZ against the reference ZoBell NosZ were ≥59.8. In addition, the predicted structures for all clade I NosZ showed the root mean square deviation (RMSD) value of <2.0 and had no structurally dissimilar amino acid residues of longer than 80 amino acids (aa) to the reference NosZ. In contrast, the Z scores for the NosZ of *D. aromatica* and *G. aurantiaca* (clade II) were 50.8 and 49.3, respectively. Poor matches with the query sequence were obtained for the clade II NosZ with RMSD values of >2.0 and structurally dissimilar amino acid residues of >80 aa. Most of the structural heterogeneity was observed in the C and N terminals.

## DISCUSSION

Biological N$_2$O reduction is generally believed to occur under strictly anaerobic conditions. The oxygen sensitivity of N$_2$O reduction can be explained by (i) the transcriptional regulation of *nos* and (ii) the inactivation of N$_2$OR by molecular oxygen. The transcription of *nosZ* can be regulated directly or indirectly by O$_2$-sensing transcriptional regulators. For instance, the transcription of *nosZ* is directly regulated by fumarate and nitrate reductase protein (FnrP) in response to oxygen depletion in *P. denitrificans* (19). *P. aeruginosa* also has similar FNR-type sensing regulators; the cascading regulation of anaerobic regulator of arginine deiminase and nitrate reductase (ANR) and dissimilatory nitrate respiration regulator (DNR) indirectly controls the synthesis of N$_2$OR (20). Another potential explanation of oxygen sensitivity points to the inactivation of N$_2$OR upon exposure to oxygen. The N$_2$OR isolated under aerobic and anaerobic conditions exhibited various redox and spin states of copper in active sites. Under limited exposure to oxygen, the enzyme shifted in electron paramagnetic resonance spectra but retained its N$_2$O-reducing activity (21). In contrast, aerobic incubation caused loss of copper content and inactivation of the catalytic site. Inactivation of N$_2$OR by oxygen was also reported due to irreversible confirmation changes. A sulfur atom binding to the active site of N$_2$OR isolated from *S. stutzeri* ZoBell was lost during aerobic enzyme isolation, leading to irreversible inactivation (6).

However, these two mechanisms do not explain the occurrence of O$_2$-tolerant N$_2$O reducers. The cells used for the microsensor experiments were incubated under anaerobic conditions with the addition of nitrite or N$_2$O to induce the expression of N$_2$OR. One exception is for *G. aurantiaca* T-27$^T$. This strain was incubated under aerobic conditions, as *G. aurantiaca* T-27$^T$ is an obligate aerobic bacterium that can express *nosZ* in the presence of O$_2$ (11, 22). The same cell cultures were used for aerobic and anaerobic N$_2$O reduction rate measurements; therefore, the initial level of N$_2$OR expressed in the cells should be the same between the two conditions (i.e., aerobic versus anaerobic N$_2$O reduction). Consequently, the transcriptional regulation of *nos* is not contributing to the O$_2$ tolerance during N$_2$O reduction of each tested strain.

In addition, the structures of NosZ, including the active sites, were highly similar between O$_2$-tolerant, -sensitive, and -intolerant N$_2$O reducers. Based on the structural similarity and the presence of conserved residues in the active sites, all of the active sites of NosZ and copper cofactors examined most likely receive similar inhibitory effects upon exposure to oxygen (6, 21). Despite similar N$_2$O respiration and bioenergetics in clade I and clade II NosZ, other accessory proteins encoded by the *nos* cluster are expected to function differently (23). These auxiliary processes could be involved in the maintenance and repair of NosZ, with detailed mechanisms remaining unclear.

Another mechanism that may explain the observed occurrence of O$_2$-tolerant N$_2$O reduction is the scavenging of O$_2$ in the cells. A whole-cell assay (as opposed to the assay done with isolated enzymes) was used in this study to calculate the N$_2$O and O$_2$ consumption rates. When both N$_2$O and O$_2$ are present, facultatively anaerobic bacteria (e.g., denitrifiers) usually prioritize the respiration of O$_2$ over N$_2$O because aerobic respiration is more favorable from both bioenergetic and kinetic perspectives (24). A rapid O$_2$ consumption rate can potentially lower the *in situ* O$_2$ concentration in the periplasm, where N$_2$OR is located. From a simplified estimation shown in the supplemental materials, an O$_2$ consumption rate of 1 $\mu$M/s/OD can cause a significant decrease in O$_2$ concentration across cell membranes. When the O$_2$ respiration rate is comparable to the O$_2$ diffusion rate that replenishes dissolved oxygen in the periplasm, the local oxygen minimum could protect N$_2$OR from inhibition in O$_2$-tolerant N$_2$O reducers. From the tested strains, we indeed observed that bacteria with higher oxygen consumption rates generally have greater oxygen tolerances (Fig. 3B). A threshold of O$_2$ consumption rate could potentially exist, where a lower rate could not emulate the diffusion rate of O$_2$ sustaining an anaerobic zone for N$_2$OR. Such a protection mechanism could be analogous to the respiration of O$_2$ in *Azotobacter* protecting O$_2$-sensitive nitrogenase (25).

Our results have some implications for N$_2$O removal applications. N$_2$O-reducing bacteria, including some of the strains examined in this study, have been used for N$_2$O mitigation in natural and engineered systems (2). For instance, bioaugmentation of *S. stutzeri* TR2 to denitrifying activated sludge has been demonstrated to mitigate N$_2$O emissions (26, 27). *Azospirillum brasilense* strains were also used as a microbial inoculant for N$_2$O mitigation in soil (28). Nevertheless, engineering applications of biological N$_2$O mitigation face major challenges, including the oxygen sensitivity of N$_2$O reduction due to the coexistence and fluctuations of dissolved oxygen and N$_2$O concentrations commonly observed in natural and engineered systems. Based on the classification of O$_2$ tolerance in this study, kinetic parameters can be used as selection criteria for microorganisms in environmental applications. Oxygen-tolerant N$_2$ORs were identified only in *S. stutzeri* in this study. *S. stutzeri* also exhibited some interesting kinetics when both electron acceptors (O$_2$ and N$_2$O) are present. The TR2 strain showed preferred N$_2$O respiration over oxygen, contrary to predictions based on electron supply rate to the electron transport chain (29). In addition, the ZoBell strain can reduce N$_2$O fast and in the presence of O$_2$, making it promising for N$_2$O bioremediation applications. Besides N$_2$O reduction rates, microorganisms with low $K_m$ values, such as *P. denitrificans* and *D. aromatica*, could be useful in scavenging low concentrations of dissolved N$_2$O. It is important to note, however, that the kinetics and O$_2$ sensitivity of N$_2$O reducers can be influenced by environmental factors, such as the type of organic carbons (30) and temperature (31). Therefore, when selecting appropriate N$_2$O reducers

for engineering applications, their N$_2$O reduction kinetics and O$_2$ sensitivity should be measured under environmentally relevant conditions.

## MATERIALS AND METHODS

**Bacterial strains.** *Stutzerimonas stutzeri* strain TR2 was kindly provided by Otsubo, Miyauchi, and Endo at Tohoku Gakuin University, Japan. *Stutzerimonas stutzeri* strain ZoBell (=ATCC 14405) and *Dechloromonas aromatica* strain RCB (=ATCC BAA-1848) were obtained from the American Type Culture Collection (ATCC). *Pseudomonas aeruginosa* PAO1 (=JCM 14847), *P. denitrificans* JCM 21484$^T$, and *A. brasilense* Sp7$^T$ (=JCM 1224$^T$) were obtained from the Japan Collection of Microorganisms (JCM). *Gemmatimonas aurantiaca* T-27$^T$ (=NBRC 100505$^T$) was obtained from Biological Resource Center (NBRC; Kisarazu, Japan).

These strains, except for *D. aromatica* RCB and *G. aurantiaca* T-27$^T$, were grown on R2A agar plates amended with 10 mM acetate and 5 mM nitrite under aerobic conditions. After 48 h of incubation at 30℃, single colonies were picked and transferred to 10 mL of R2A broth with 10 mM acetate and 5 mM nitrite. Each liquid culture was incubated in a sealed tube with an N$_2$ atmosphere at 30℃ until harvested during the exponential growth phase. *D. aromatica* RCB was grown on Trypticase soy agar (TSA) supplemented with 5% defibrinated sheep blood under anaerobic conditions at 30℃ for 10 days. Single colonies were transferred to 10 mL of R2A broth supplemented with 20 mM lactate and incubated under a 1.39% N$_2$O atmosphere (in N$_2$) at 30℃ until harvested. *G. aurantiaca* T-27$^T$ was grown on R2A agar under aerobic conditions. Single colonies were transferred to 10 mL of R2A broth and aerobically incubated at 25℃ until harvested. The addition of nitrite inhibited the growth of *G. aurantiaca*, which was expected to have an incomplete denitrification pathway (32).

**Microsensor experiments.** Bacterial cultures were harvested during the early to mid-exponential growth phase as determined by the optical density at 600 nm (OD$_{600}$) measurement. Cultures were washed twice with a sterile 10 mM piperazine-N,N′-bis(2-ethanesulfonic acid) (PIPES) buffer (pH 7.5) and resuspended in a PIPES buffer supplemented with 10 mM sodium acetate. The cell suspensions were purged with a gas mix of N$_2$O (1.39%, vol/vol) in N$_2$ for 10 min to achieve targeted levels of dissolved N$_2$O concentrations (300 $\mu$M). The cell suspensions were then diluted with PIPES buffer to the desired concentration (~10$^6$ CFU/mL; OD$_{600}$, ~0.1) and transferred to a double chamber containing mini stirrer bars (Unisense, Aarhus, Denmark) (see Fig. S5 in the supplemental material). The chamber was capped and placed in a sensor rack with built-in stirrers and submerged in a 30℃ water bath. An N$_2$O microsensor and an O$_2$ microsensor (Unisense) were inserted into the chamber via small halls to measure dissolved N$_2$O and O$_2$ concentrations every second for up to 2 h or until O$_2$ depletion. The N$_2$O and O$_2$ microsensors were two-point calibrated with zero and saturated solutions (300 $\mu$M for N$_2$O and 236 $\mu$M for O$_2$) at 30℃. No cross interference was observed between N$_2$O and O$_2$ on respective microsensors (see Table S1 in the supplemental material). The OD$_{600}$ of the cell suspension was recorded at the end of each microsensor test. At least three independent microsensor measurements were done for each strain.

The measured concentrations of N$_2$O and O$_2$ were averaged over time intervals of 100 to 1000 s depending on the duration of microsensor tests. This step is useful to minimize the noise generated by the microsensors. Linear rates for N$_2$O consumption were extrapolated within each time interval. The Michaelis-Menten plots were then constructed using the rates and corresponding N$_2$O concentrations. A nonlinear least square method with the Levenberg-Marquardt algorithm (33) was used for curve fitting on Origin 2021 (version 9.8.0.200) to determine kinetic parameters, including the maximum rate ($V_{max}$) and the Michaelis constant ($K_m$). Similarly, $V_{max}$ for O$_2$ was linearly extrapolated from O$_2$ concentrations measured by the microsensor.

**Bioinformatics and comparative protein structure modeling.** The NosZ sequences of the selected strains (GenBank accession numbers WP_011287329, EHY76008, BAM68548, NP_252082, QEL93987, WP_156798935, and Q51705 for *D. aromatica* RCB, *S. stutzeri* ZoBell, *S. stutzeri* TR2, *P. aeruginosa* PAO1, *A. brasilense* Sp7, *G. aurantiaca* T-27, and *P. denitrificans* JCM 21484, respectively) were retrieved from National Center for Biotechnology Information (NCBI; https://www.ncbi.nlm.nih.gov). Multiple sequence alignment and phylogenetic tree construction were done using the neighbor-joining method without distance correction by using Clustal Omega (https://www.ebi.ac.uk/Tools/msa/clustalo/). NosZ structures were predicted through the nondocker implementation of AlphaFold2 version 2.1.1 via the Minnesota Supercomputing Institute (MSI). The NosZ sequence of the selected strains was used as the input with the default prediction parameters to run on a Linux environment. The best-predicted protein models were selected for each sequence and loaded into PyMOL (Schrödinger, Inc., New York, NY). All models were colored based on their predicted local distance difference test (pLDDT) that are stored in the B-factor fields of the PDB files. All predicted structures were compared against each other using DaliLite.v5 (http://ekhidna2.biocenter.helsinki.fi/dali) (34).

## SUPPLEMENTAL MATERIAL

Supplemental material is available online only.

**SUPPLEMENTAL FILE 1**, PDF file, 5.5 MB.

## ACKNOWLEDGMENTS

We thank Wakako Otsubo, Keisuke Miyauchi, and Ginro Endo Tohoku Gakuin University, Japan for providing *Stutzerimonas stutzeri* strain TR2. We also thank Sujin Yeom and Mike Blazanin for their help with the initial experimental setup and Carrie Wilmot for valuable comments.

This work was supported by the Biocatalysis program and the MnDRIVE Initiative of the University of Minnesota.

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
