## [Reviewer comments · Microbiology Spectrum]

Microbiology Spectrum

Clarifying Microbial Nitrous Oxide Reduction Under Aerobic Conditions: Tolerant, Intolerant, and Sensitive

Zhiyue Wang, Nisha Vishwanathan, Sophie Kowaliczko, and Satoshi Ishii

Corresponding Author(s): Satoshi Ishii, University of Minnesota Twin Cities

Review Timeline:

Submission Date:	November 17, 2022
Editorial Decision:	February 6, 2023
Revision Received:	February 8, 2023
Accepted:	February 18, 2023

Editor: Victor Gonzalez

Reviewer(s): Disclosure of reviewer identity is with reference to reviewer comments included in decision letter(s). The following individuals involved in review of your submission have agreed to reveal their identity: Maria J Delgado (Reviewer #1)

Transaction Report:

DOI: <https://doi.org/10.1128/spectrum.04709-22>

February 6, 2023

Prof. Satoshi Ishii
University of Minnesota Twin Cities
BioTechnology Institute
1479 Gortner Ave.
140 Gortner Lab
St. Paul, MN 55108

Re: Spectrum04709-22 (Clarifying Microbial Nitrous Oxide Reduction Under Aerobic Conditions: Tolerant, Intolerant, and Sensitive)

Dear Prof. Satoshi Ishii:

Link Not Available

Sincerely,

Victor Gonzalez

Journals Department
Reviewer comments:

Reviewer #1 (Comments for the Author):

Minor Comments to the authors

General comment. In the results section concerning to the explanation of figures 1 an 2, there is not reference in the text to figures A, B, C, and D. Thus, this make difficult the conexion between the figures and the information included in the results text resulting in a dificult comprehension of the message.

Reviewer #2 (Comments for the Author):

This paper presents some very interesting data concerning the tolerance of different N₂O reducing microorganisms to oxygen. The authors also extend the comparison, perhaps too far in my opinion, to include statements about Clade I and Clade II NosZ. The authors identify two *Stutzerimonas* strains as being clearly O₂ tolerant when reducing N₂O, while other strains with both Clade I and Clade II NosZ are O₂ intolerant. The remaining strains tested were classified as O₂ sensitive, although I would argue that a special case of sensitivity could be applied to *Gemmatimonas*. The work is interesting and has potential applications in attenuating N₂O being produced in different ecosystems. However, I have identified some issues that I think need to be addressed to make the paper stronger.

1. I think care must be taken when stating any conclusions about comparing how Clade I and Clade II NosZ containing organisms will respond in terms of O₂. The authors only evaluated two Clade II NosZ organisms, one of which only expresses NosZ after being grown aerobically. Clade II NosZ exhibits considerably more divergence than Clade I NosZ (At least 8 different subclades of Clade II can be clearly delineated). It is therefore not appropriate to make statements about how microorganisms with Clade II NosZ will respond in the presence of oxygen. I would recommend the authors reviewing their text and making sure that such statements are avoided.
2. The authors should in the Introduction make sure to acknowledge previous work done with *Gemmatimonas* N₂O reduction and O₂. They do not cite the following paper: Chee-Sanford, J., Tian, D. & Sanford, R. Consumption of N₂O and other N-cycle intermediates by *Gemmatimonas aurantiaca* strain T-27. *Microbiology* 165, 1345-1354 (2019). Reference 21 also contains some information about O₂, however the authors do not mention this in the Introduction.
3. Ln 90-91: The authors state that *Gemmatimonas* could not reduce N₂O in the presence of O₂, however this is inconsistent with what they state below and what is shown in Figure 2-C2. It seems this organism's rate of N₂O reduction increases as the O₂ is depleted. This is consistent with the data reported in the Chee-Sanford et. al reference.
4. Ln 140-141: What does "highly similar" mean? Clearly this strain behaves different than the *Stutzerimonas* strains in terms of O₂ sensitivity to N₂O reduction. Is the ~80% similarity really "highly similar". I am not sure I see the point being made here.
5. Ln 193-195: Which *St. stutzeri* strain was used here? It is confusing to state this because your results do not seem to indicate this occurs with the cultures you tested of *St. stutzeri*.
6. Figure 3: The caption mentions V_{max} for N₂O reduction under aerobic and anaerobic conditions and for O₂ consumption, however the axis labels on both A and B panels are not very clear. I would recommend labeling (max N₂O reduction rate) to make it clear what is being represented.
7. What is interesting from the data and not discussed here is that the *Stutzerimonas* strains actually have greater maximum N₂O reduction rates than maximum O₂ reduction rates (Figure 3b). Also Figure 3a seems to show that strain Zobell actually has a higher maximum N₂O reduction rate under aerobic conditions than under anaerobic conditions. This is interesting and brings up the question are these organisms gaining energy for growth from the N₂O reduction? It seems the authors do not even consider this, and instead only focus on the sensitivity of the activity to O₂. Why do *Stutzerimonas* strains do this? In the presence of O₂, they are diverting at least an equivalent portion of electrons to reducing N₂O. In the case of strain TR2, it is even larger (almost a 2:1 ratio) .I would suggest that the authors consider adding some discussion about this.

Staff Comments:

Preparing Revision Guidelines

- Point-by-point responses to the issues raised by the reviewers in a file named "Response to Reviewers," NOT IN YOUR COVER LETTER.
- Upload a compare copy of the manuscript (without figures) as a "Marked-Up Manuscript" file.

- Each figure must be uploaded as a separate file, and any multipanel figures must be assembled into one file.
- Manuscript: A .DOC version of the revised manuscript
- Figures: Editable, high-resolution, individual figure files are required at revision, TIFF or EPS files are preferred

Please return the manuscript within 60 days; if you cannot complete the modification within this time period, please contact me. If you do not wish to modify the manuscript and prefer to submit it to another journal, please notify me of your decision immediately so that the manuscript may be formally withdrawn from consideration by Microbiology Spectrum.

Response to Reviewers

Reviewer #1

General comment. In the results section concerning to the explanation of figures 1 and 2, there is not reference in the text to figures A, B, C, and D. Thus, this make difficult the connection between the figures and the information included in the results text resulting in a difficult comprehension of the message.

We appreciate the reviewer's suggestion on improving the readability of the result section. We have added text references to each panel of the figures. Lines 79-97 now read as follows.

“Under anaerobic conditions, *St. stutzeri* TR2 (Clade I N₂OR, **Figure 1-B1**) had the highest V_{\max} ($8.37 \pm 0.81 \mu\text{M/s/OD}$), whereas *G. aurantiaca* T-27 (Clade II N₂OR, **Figure 2-C1**) had the lowest V_{\max} ($0.13 \pm 0.02 \mu\text{M/s/OD}$). This general trend in kinetics, however, may not extend beyond the studied strains, especially given the diversity of microorganisms harboring Clade II NosZ (12).

The ability to reduce N₂O in the presence of O₂ varied by strain, and there was no overall trend between the tested strains with Clade I and II N₂OR. For example, *St. stutzeri* TR2 (**Figure 1-B2**) reduced N₂O under aerobic conditions with the V_{\max} of $6.65 \pm 0.37 \mu\text{M/s/OD}$, whereas *Ps. aeruginosa* PAO1 (Clade I N₂OR, **Figure 2-A1**) could not reduce N₂O in the presence of O₂. *D. aromatica* RCB (Clade II N₂OR, **Figure 2-B1**) also could not reduce N₂O in the presence of O₂. In contrast, *G. aurantiaca* T-27 (Clade II N₂OR, **Figure 2-C2**) exhibited a very slow N₂O reduction rate under aerobic conditions, which increased once O₂ was depleted. *Azospirillum brasilense* Sp7 (Clade I N₂OR, **Figure 1-D2**) reduced N₂O in the presence of O₂ up to 180 μM ; however, its V_{\max} could not be fitted to the Michaelis-Menten model.”

Reviewer #2

This paper presents some very interesting data concerning the tolerance of different N₂O reducing microorganisms to oxygen. The authors also extend the comparison, perhaps too far in my opinion, to include statements about Clade I and Clade II NosZ. The authors identify two *Stutzerimonas* strains as being clearly O₂ tolerant when reducing N₂O, while other strains with both Clade I and Clade II NosZ are O₂ intolerant. The remaining strains tested were classified as O₂ sensitive, although I would argue that a special case of sensitivity could be applied to *Gemmatimonas*. The work is interesting and has potential applications in attenuating N₂O being produced in different ecosystems. However, I have identified some issues that I think need to be addressed to make the paper stronger.

We appreciate that the reviewer recognized the novelty and significance of this study, and we agree that accurate statements and careful generalization of results are critical. Thus, we have checked for any overstatement regarding the general comparison of Clade I and Clade II traits throughout the original manuscript and revised accordingly. We have the following point-by-point responses below.

1. I think care must be taken when stating any conclusions about comparing how Clade I and Clade II NosZ containing organisms will respond in terms of O₂. The authors only evaluated two Clade II NosZ organisms, one of which only expresses NosZ after being

grown aerobically. Clade II NosZ exhibits considerably more divergence than Clade I NosZ (At least 8 different subclades of Clade II can be clearly delineated). It is therefore not appropriate to make statements about how microorganisms with Clade II NosZ will respond in the presence of oxygen. I would recommend the authors reviewing their text and making sure that such statements are avoided.

We agree with the reviewer that the tested strains with Clade II NosZ are limited in this study and we should avoid overgeneralization of comparative findings. We have checked the entire manuscript. We also added a hedging statement on Line 84-86 as follows.

“This general trend in kinetics, however, may not extend beyond the studied strains, especially given the diversity of microorganisms harboring Clade II NosZ.”

2. The authors should in the Introduction make sure to acknowledge previous work done with *Gemmatimonas* N₂O reduction and O₂. They do not cite the following paper: Chee-Sanford, J., Tian, D. & Sanford, R. Consumption of N₂O and other N-cycle intermediates by *Gemmatimonas aurantiaca* strain T-27. Microbiology 165, 1345-1354 (2019). Reference 21 also contains some information about O₂, however the authors do not mention this in the Introduction.

We agree with the reviewer and have added this reference (Ref. #11) to the introduction (L51) and discussion sections (L202) in addition to Ref. #21.

3. Ln 90-91: The authors state that *Gemmatimonas* could not reduce N₂O in the presence of O₂, however this is inconsistent with what they state below and what is shown in Figure 2-C2. It seems this organism's rate of N₂O reduction increases as the O₂ is depleted. This is consistent with the data reported in the Chee-Sanford et. al reference.

We agree with the reviewer that the original statement about *Gemmatimonas* is inaccurate. The revised statement (Line 93-95) now read as follows.

“In contrast, *G. aurantiaca* T-27 (Clade II N₂OR, **Figure 2-C2**) exhibited a very slow N₂O reduction rate under aerobic conditions, which increased once O₂ was depleted.”

4. Ln 140-141: What does "highly similar" mean? Clearly this strain behaves different than the *Stutzerimonas* strains in terms of O₂ sensitivity to N₂O reduction. Is the ~80% similarity really "highly similar". I am not sure I see the point being made here.

We are refereeing to the high similarity of amino acid sequences. An 80% similarity is relatively high when comparing amino acid sequences between different species. We clarified the description in the revised sentence on Line 142-145.

“However, *Ps. aeruginosa* PAO1, which showed oxygen-intolerant N₂O reduction, also has similar NosZ amino acid sequences to *St. stutzeri* (77.5% with the ZoBell strain and 79.7% with the TR2 strain).”

5. Ln 193-195: Which *St. stutzeri* strain was used here? It is confusing to state this because your results do not seem to indicate this occurs with the cultures you tested of *St. stutzeri*.

Wüst et al. (Ref. #6) was referring to the N2OR of *St. stutzeri* ZoBell, the same strain as this study. However, it is based on purified enzyme instead of whole cell studies. We have removed the term “*in vivo*” to avoid confusion. Line 194-196 now read as follows.

“A sulfur atom binding to the active site of N2OR isolated from *St. stutzeri* ZoBell was lost during aerobic enzyme isolation, leading to irreversible inactivation [6].”

6. Figure 3: The caption mentions V_{\max} for N_2O reduction under aerobic and anaerobic conditions and for O_2 consumption, however the axis labels on both A and B panels are not very clear. I would recommend labeling (max N_2O reduction rate) to make it clear what is being represented.

We agree with the reviewer and have changed the axis labels on Figure 3 as suggested by the reviewer.

7. What is interesting from the data and not discussed here is that the *Stutzerimonas* strains actually have greater maximum N_2O reduction rates than maximum O_2 reduction rates (Figure 3b). Also Figure 3a seems to show that strain ZoBell actually has a higher maximum N_2O reduction rate under aerobic conditions than under anaerobic conditions. This is interesting and brings up the question are these organisms gaining energy for growth from the N_2O reduction? It seems the authors do not even consider this, and instead only focus on the sensitivity of the activity to O_2 . Why do *Stutzerimonas* strains do this? In the presence of O_2 , they are diverting at least an equivalent portion of electrons to reducing N_2O . In the case of strain TR2, it is even larger (almost a 2:1 ratio). I would suggest that the authors consider adding some discussion about this.

We appreciate the reviewer’s suggestive comment. TR2 strain indeed showed a preference for nitrous oxide in electron transport ($4 e^-$ per mole of O_2 vs. $2 e^-$ per mole of N_2O). We addressed the unique kinetics observed in *St. stutzeri* strains by citing Ref. 29 on Line 248 – 251 as follows.

“*St. stutzeri* also exhibited some interesting kinetics when both electron acceptors (O_2 and N_2O) are present. TR2 strain showed preferred N_2O respiration over oxygen, contrary to predictions based on electron supply rate to the electron transport chain [29].”

February 18, 2023

Prof. Satoshi Ishii
University of Minnesota Twin Cities
BioTechnology Institute
1479 Gortner Ave.
140 Gortner Lab
St. Paul, MN 55108

Re: Spectrum04709-22R1 (Clarifying Microbial Nitrous Oxide Reduction Under Aerobic Conditions: Tolerant, Intolerant, and Sensitive)

Dear Prof. Satoshi Ishii:

Your manuscript has been accepted, and I am forwarding it to the ASM Journals Department for publication. You will be notified when your proofs are ready to be viewed.

Sincerely,

Victor Gonzalez
Editor, Microbiology Spectrum
